# cfDNA and DNases: New Biomarkers of Sepsis in Preterm Neonates—A Pilot Study

**DOI:** 10.3390/cells11020192

**Published:** 2022-01-06

**Authors:** Moritz Lenz, Thomas Maiberger, Lina Armbrust, Antonia Kiwit, Axel Von der Wense, Konrad Reinshagen, Julia Elrod, Michael Boettcher

**Affiliations:** 1Department of Pediatric Surgery, University Medical Center Hamburg-Eppendorf, 20246 Hamburg, Germany; moritz.lenz@stud.uke.uni-hamburg.de (M.L.); lina.armbrust@hotmail.de (L.A.); 7080305@stud.uke.uni-hamburg.de (A.K.); k.reinshagen@uke.de (K.R.); julia.elrod@umm.de (J.E.); 2Department of Pediatric Intensive Care and Neonatology, Altona Children’s Hospital, 22763 Hamburg, Germany; thomas.maiberger@kinderkrankenhaus.net (T.M.); axel.wense@kinderkrankenhaus.net (A.V.d.W.); 3Department of Pediatric Surgery, University Medical Center Mannheim, Heidelberg University, 68167 Mannheim, Germany

**Keywords:** sepsis, NETs, extracellular DNA, neutrophil extracellular traps, preterm infants, early onset neonatal sepsis, late onset neonatal sepsis

## Abstract

Introduction: An early and accurate diagnosis of early onset neonatal sepsis (EONS) and late onset neonatal sepsis (LONS) is essential to improve the outcome of this devastating conditions. Especially, preterm infants are at risk. Reliable biomarkers are rare, clinical decision-making depends on clinical appearance and multiple laboratory findings. Markers of NET formation and NET turnover might improve diagnostic precision. Aim of this study was to evaluate the diagnostic value of NETs in sepsis diagnosis in neonatal preterm infants. Methods: Plasma samples of neonatal preterm infants with suspected sepsis were collected. Blood samples were assayed for markers of NET formation and NET turnover: cfDNA, DNase1, nucleosome, NE, and H3Cit. All clinical findings, values of laboratory markers, and epidemiological characteristics were collected retrospectively. Two subpopulations were created to divide EONS from LONS. EMA sepsis criteria for neonatal sepsis were used to generate a sepsis group (EMA positive) and a control group (EMA negative). Results: A total of 31 preterm neonates with suspected sepsis were included. Out of these, nine patients met the criteria for sepsis according to EMA. Regarding early onset neonatal sepsis (3 EONS vs. 10 controls), cfDNA, DNase I, nucleosome, and CRP were elevated significantly. H3Cit and NE did not show any significant elevations. In the late onset sepsis collective (6 LONS vs. 12 controls), cfDNA, DNase I, and CRP differed significantly compared to control group.

## 1. Introduction

Neonatal sepsis remains a life-threatening medical condition in the postnatal period [1] with an incidence of 2.2/1000 live births in middle- and high-income countries [2]. Neonatal sepsis is subdivided into two entities [3]. One is early onset neonatal sepsis (EONS), which occurs within the first 72 h after birth and usually results from acquisition of maternal microorganisms. The case fatality rate is around 16%, however, in preterm neonates born at 22–24 weeks of gestational age, mortality rate is 54% [1]. In preterm infants, infections with *Escherichia coli* dominate the microbiological spectrum with 81% [1]. The second entity is late onset neonatal sepsis (LONS), which occurs later than 72 h after birth, and is caused by microorganisms from the environment [3]. Vascular-access catheters, mechanical ventilation and other nosocomial interventions, which are more commonly required during intensive care of preterm infants, are risk factors for LONS [1]. In both entities, empirical antibiotic treatment should be started immediately to improve prognosis, without waiting for pathogen detection and identification [1]. Overall mortality rate of LONS is around 13% and increases with low birth weight and prematurity [4].

Symptoms of neonatal sepsis are non-specific and therefore vital parameters and symptoms of infection must be observed thoroughly to identify sepsis as soon as possible [5]. Established laboratory markers, such as C-reactive protein (CRP), must be considered for diagnosis finding and therapeutical decision-making. European Medical Agency (EMA) developed a scoring system to find an operational sepsis definition in 2010 [6]. Detailed information about the clinical and laboratory parameters included in the EMA sepsis scoring system can be found in Table 1. Nevertheless, there is still a lack of sensitive and reliable biomarkers for sepsis diagnosis, especially in neonates [5,7].

In the first days of life, the immune system is faced with a broad spectrum of microbes and pathogens for the first time. As a crucial part of the innate immune system especially in neonates, neutrophils represent the first defense line against them [1,9]. This subpopulation of leucocytes can attack and immobilize pathogens by releasing neutrophil extracellular traps (NETs) [9,10]. NETs consist of DNA filaments, citrullinated histones, and proteolytic enzymes, such as neutrophil elastase (NE) [11]. Degradation of NETs by deoxyribonucleases (DNases) ensures homeostasis of the immune response [10]. The cytokine storm during septic shock may lead to an imbalance of NET formation and NET degradation, resulting in complications like thrombotic events and tissue damage [12,13,14]. Elevated markers of NET formation are described in the context of sepsis and other inflammatory diseases [13]. Aim of this study was to evaluate the value of markers of NET formation and turnover for sepsis diagnosis in a collective of preterm neonates.

## 2. Materials and Methods

### 2.1. Study Design

Leftovers of blood samples for diagnostic purposes from neonatal patients were collected prospectively at the neonatal intensive care unit, Altona Children’s Hospital, Hamburg, Germany from May 2020 to August 2021. A parent’s informed written consent was obtained prior to blood collection. This trial meets the guidelines of the medical research ethics committee of Hamburg (Ethik-Kommission der Ärztekammer Hamburg, PV4991) and the Helsinki Declaration of 1964. The study was preregistered at ClinicalTrials.gov (NCT02567305). All clinical data were collected retrospectively and anonymously by two independent investigators.

According to WHO definition for preterm infants [15], only neonates with a gestational age of less than 37 completed weeks were included. All neonates with suspected sepsis were eligible for participation independent from other secondary diagnosis. However, only blood samples collected before antibiotic therapy were included. The study cohort was divided into two groups based on the age at time of sample collection; 3 days of age was defined as a cut-off value to differentiate EONS and LONS [3]. To evaluate the occurrence of clinical sepsis, European Medical Agency (EMA) sepsis scoring system [6] was used. A positive EMA sepsis score (at least two positive clinical and two positive laboratory categories, and a suspected or proven infection) were needed to assign a patient to the sepsis group. All other individuals were defined as control group. A flow diagram for recruitment and subdivision of the study cohort can be found in Figure 1.

Plasma samples were assayed for markers of NET formation and NET degradation, including circulating free deoxyribonucleic acid (cfDNA), nucleosome, deoxyribonuclease I (DNase I), citrullinated Histone 3 (H3Cit), and neutrophil elastase (NE).

### 2.2. Sample Preparation

Immediately after sample collection, the collected EDTA blood leftovers were centrifuged (2000rcf, 21 °C, 10 min), the resulting plasma was aliquoted and stored at −80 °C until assays were performed.

### 2.3. Circulating Free Deoxyribonucleic Acid (cfDNA)

Concentration of cfDNA was measured with an established fluorescence-based assay using Sytox Orange Nucleic Acid Stain (Invitrogen, Eugene, OR, USA), as described previously [16]. Briefly, a DNA standard curve (range 0–2000 ng/mL) was generated by serial dilution and placed on a 96-well microtiter plate; 50 µL of each plasma sample (dilution 1:20) were placed four times on the plate. Two wells each were incubated with a Sytox Orange solution. As a blank value, the other two wells were treated with a dilution buffer (0.1% BSA, 2mM EDTA in PBS) only. Fluorescence measurement (Ex: 544 nm, Em: 570 nm) was carried out after 5 min. Results are expressed in ng/mL.

### 2.4. Nucleosome

Plasma nucleosome level was assayed using a modified Cell Death Detection ELISA kit (Roche, Rotkreuz, Switzerland). In order to obtain a comparable standard curve, the included positive control was diluted 1:5, 1:10, 1:20, 1:40, 1:80, 1:160, 1:320 and 1:640. All other steps were carried out according to the manufacturer’s protocol. Optical density values were interpolated generating a four-parameter logistic (4-PL) curve-fit. Results are expressed in relative units on a nondimensional scale (0–1000).

### 2.5. Deoxyribonuclease I (DNase I) ELISA

DNase I, which plays a crucial role in NET turnover [10], was assayed using the Human DNase I ELISA Kit (MyBioSource, San Diego, CA, USA). Producer’s instructions were followed and results are expressed in ng/mL. 

### 2.6. Neutrophile Elastase (NE) ELISA

Neutrophil Elastase, a marker for NETosis, was measured using the Human Polymorphonuclear (PMN)-Elastase ELISA Kit (Invitrogen, Thermo Fisher Scientific, Waltham, MA, USA). The manufacturer’s protocol was followed, and results are expressed in ng/mL.

### 2.7. Citrullinated Histone 3 (H3Cit) ELISA 

The concentration of H3Cit, which is an integral component of NETs [10], was assayed using the Citrullinated Histone H3 (Clone 11D3) ELISA Kit (Cayman Chemicals, Ann Arbor, MI, USA). All manufacturer’s instructions were followed.

### 2.8. Statistics

All data were analyzed using SPSS Statistics 26 (IBM, Armonk, NY, USA) and GraphPad Prism 9 (GraphPad, San Diego, CA, USA). As this was a pilot study, power estimation was deduced from previous trials regarding sepsis in neonates [17]. Differences between groups were calculated using Mann–Whitney tests. The data were illustrated with violin plots including markers for mean and interquartile range. The level of significance was set at 0.05. Tables of clinical characteristics were calculated using Mann–Whitney tests and Fisher’s exact tests. Level of significance was set at 0.05.

## 3. Results

A total of 31 neonates with suspected sepsis were included. Of these, nine patients met the criteria for sepsis according to EMA. The study collective was subdivided into the two entities of EONS and LONS. Clinical characteristics of the two study cohorts are tabularized in Table 2 for EONS and Table 3 for LONS, respectively. In both study groups, personal characteristics like age, gender, and gestational age did not differ significantly. Mortality rate was only significantly elevated in EONS group (*p* = 0.038). Only one patient from LONS group had a proven infection (via microbiological culture of a peripheral blood sample). Interestingly, we found no significant differences in leucocyte count in both entities, but a significant thrombocytopenia (EONS: *p* = 0.012; LONS: *p* = 0.006). Regarding blood gas analysis, lactate was elevated in EONS patients (*p* = 0.023). EONS individuals showed a significantly elevated rate of hyperglycemia (*p* = 0.018). Individual clinical information and scoring results of all EMA positive participants can be found in Table 4.

As shown in Figure 2, markers of neutrophil extracellular traps formation (cfDNA) and NET degradation (nucleosome and DNase I) were significantly elevated in EONS; as well as CRP which is one of the possible laboratory signs of sepsis according to the EMA criteria. However, NE, which is a marker of neutrophil activation, and the surrogate marker of NET formation H3Cit, did not show any significant differences.

In premature born neonates with LONS, cfDNA, and DNase I, as well as CRP were significantly elevated compared to controls as shown in Figure 3. No differences were found for nucleosome, NE, and H3Cit. 

## 4. Discussion

Early diagnosis and adequate treatment of sepsis in preterm infants remains a challenge in neonatal intensive care. Established biomarkers for neonatal sepsis are helpful for clinical decision-making but have limitations in terms of their reliability and predictive power [18]. Hence, complex scoring systems, like the sepsis criteria for neonates developed by the Pediatric Committee (PDCO) of the EMA used in this study [6], which contains clinical symptoms and laboratory signs, are necessary for diagnosis. Elevated levels of NET-associated markers have been found in septic adult patients [16], and especially cfDNA has been discussed as a biomarker for posttraumatic sepsis [19]. The current study suggests that markers of NET formation and degradation may also help to diagnose sepsis in preterm infants. 

In the present cohort of preterm neonates with EONS, blood levels of cfDNA, nucleosome, and DNase1 were elevated but no difference for the surrogate markers H3Cit and NE were found. These findings must be interpreted with caution, because of the low sample size included in the study. In a previous study, NET markers in umbilical cord blood drawn immediately after birth were unable to predict future EONS in neonates [17]. The main reason may be that prenatal NET formation is limited by the neonatal NET inhibitory factor (nNIF), which can be found in cord blood samples of neonates [20]. In neonates, nNIF seems to block pivotal steps of NET formation, such as PAD4 activity or nuclear decondensation [20]. It is assumed that this mechanism may contribute to the limited ability of very young neonates to kill bacteria, rendering them more susceptible to sepsis during the first days of life [21]. Presumably, perinatal NET formation is regulated tightly to prevent negative repercussions during perinatal adaptation, such as hyperinflammation, NET-mediated vascular injury, and thrombosis [20]. It has been reported that the effects of nNIF subside rapidly after delivery [20] and it appears that NET markers may be used to diagnose early onset sepsis in preterm neonates. However, further investigations are needed to understand the clinical relevance of NET inhibition in the first days of life of preterm infants.

In neonates with LONS, also cfDNA, DNase1, and CRP were significantly elevated, but no differences were found for NE, nucleosome and H3Cit. It stands out, that for both entities of sepsis, cfDNA and DNase1 were elevated significantly. The combinations of these two markers cover the whole process of NETosis and NET degradation [22,23]. 

Circulating free DNA is released into peripheral blood during apoptosis, necrosis, and NET formation [24,25]. CfDNA appears particularly interesting as a diagnostic biomarker because healthy individuals show minimal levels only [26,27]. High levels of cfDNA have been suggested as a predictor of mortality in adult sepsis patients [28] DNase1 regulates NETs by degradation of DNA fragments [29]. An elevated enzyme activity indicates an increased degradation of cfDNA, before cfDNA level effectively rises [22]. This effect might be an early marker for the beginning of immune-mediated dysregulation during sepsis. Interestingly, we found elevated plasma levels of DNase1 in EONS as well as in LONS group. Combined with the finding of elevated levels of cfDNA in both groups, it is possible that we found first hints for a significant influence of NETosis in sepsis evolvement in preterm infants. These impressions should be evaluated in future studies because other pathophysiological explanations are still conceivable. CfDNA and its counterpart DNase I are involved in several immunological mechanisms, like necrosis [25]. Furthermore, it would be interesting to examine the enzyme activity of DNAse1 in combination with DNase1 level to discover the protein regulation during septic conditions in preterm infants. 

As this was a pilot study with a limited number of subjects, some limitations apply. Sample size allows only a careful analysis of the role of NETs in EONS and LONS in preterm infants. Surrogate markers of NET formation like H3Cit [30] or NE [23] did not differ between sepsis and non-sepsis. 

EMA negative control group might still include sepsis patients, because EMA sepsis scoring system is only an approximation for a reliable sepsis score. However, it is also possible that other conditions like respiratory disorders resulted in elevated levels of the NET markers in the non-sepsis group. This might be a significant confounding factors since various respiratory syndromes have been associated with NETs in neonates [31]. A control group with healthy individuals only would be desirable, but hard to realize in the context of a clinical study. Recruitment on an intensive care unit is not ideal, because all patients have several medical conditions. Ethically, blood sampling in preterm neonates should be reduced to a minimum. As we used leftovers from diagnostic procedures to meet these ethnical claims, we were faced with a longer period from collection to storage at −80 °C, which might affect the quality of blood samples. To minimize this effect, we followed a strict protocol for all blood samples to make all samples comparable. 

In conclusion, surrogate markers of NET formation and degradation, especially cfDNA and DNase appear to be potential biomarkers for EONS and LONS in preterm infants. Further studies with large cohorts of preterm infants are needed to examine the real diagnostic potential of NETs. Ultimately, a combination of markers for NET formation and degradation might be promising to identify immune dysregulation in children with sepsis.

## 5. Conclusions

CfDNA and DNase appear to be potential biomarkers for diagnosis of early and late onset neonatal sepsis in preterm infants. Further studies should validate the diagnostic value of NET-associated markers in larger study collectives.

## Figures and Tables

**Figure 1 cells-11-00192-f001:**
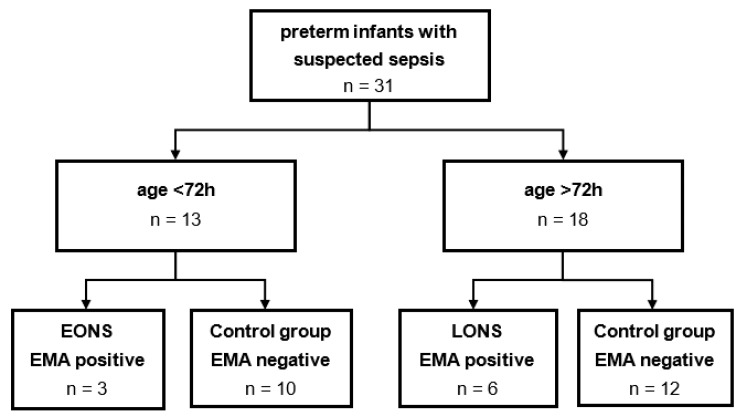
Flow diagram for study cohort subdivision. A total of 31 samples of preterm infants with suspected sepsis were collected. After the division by age at blood collection, remaining study cohorts were analyzed retrospectively for the occurrence of clinical sepsis according to EMA sepsis scoring system.

**Figure 2 cells-11-00192-f002:**
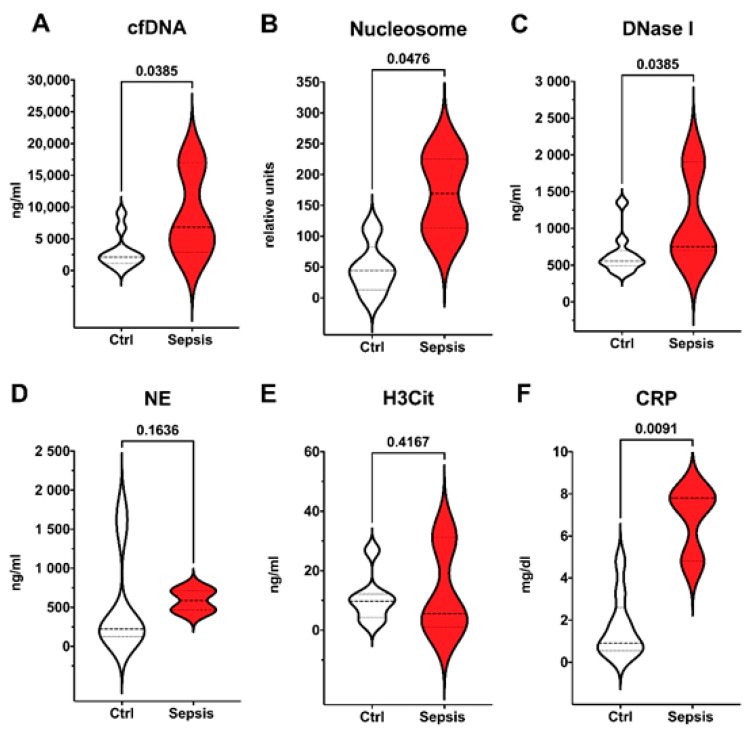
Early onset neonatal sepsis (EONS) in preterm infants. Preterm infants meeting EMA sepsis criteria and definition of EONS (red, *n* = 3) compared to control group (white, *n* = 10). Plasma levels of surrogate parameters of NET formation and degradation (**A**–**E**) and CRP (**F**) are shown. Significant differences between preterm infants were found for cfDNA (**A**); Nucleosome (**B**); DNase I (**C**), and CRP (**F**). Data are visualized with violin plots including markers for median and interquartile range. Mann–Whitney tests were used. Significance was set at 0.05.

**Figure 3 cells-11-00192-f003:**
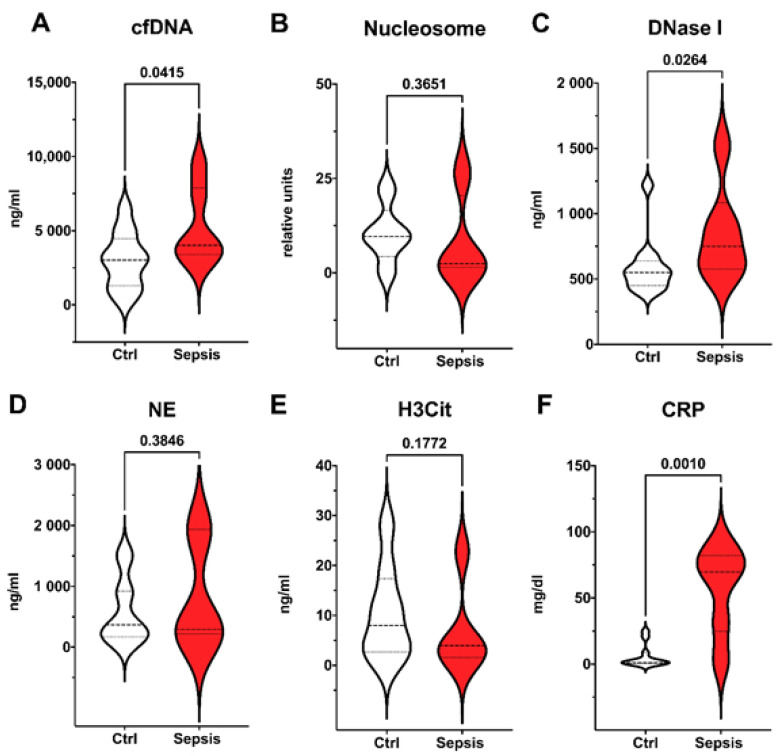
Late onset neonatal sepsis (LONS) in preterm infants. Preterm infants meeting EMA sepsis criteria and definition of LONS (red, *n* = 6) compared to control group (white, *n* = 12). Plasma levels of surrogate parameters of NETosis and NET degradation (**A**–**E**) and CRP (**F**) are shown. Significant differences were found for cfDNA (**A**); DNase I (**C**), and CRP (**F**). Data are visualized with violin plots including markers for median and interquartile range. Mann–Whitney tests were used. Significance was set at 0.05.

**Table 1 cells-11-00192-t001:** EMA sepsis scoring system, adapted from EMA publication (2010) and Odabasi and Bulbul (2020) [6,8]. The presence of at least two clinical signs and at least two laboratory findings from the table are considered as clinical sepsis in the context of a suspected or proven infection. This scoring system is suitable for EONS and LONS and is applicable up to 44 weeks of age.

EMA Sepsis Scoring System	
Clinical	Laboratory
**Body temperature:**	**Leukocyte count:**
>38.5 °C or	<4000/mm³ or >20.000/mm³
<36 °C and/or temperature irregularities	**Immature/total neutrophil ratio:**
**Cardiovascular instability:**	≥0.2
Bradycardia or tachycardia and/or	**Platelet Count:**
rhythm irregularity	<100.000/mm³
Urine amount < 1 mL/kg/h	**CRP:**
Hypotension	>15 mg/L (1.5 mg/dL)
Impaired peripheral perfusion	**Procalcitonin:**
**Skin and subcutaneous lesions:**	≥2 ng/mL
Petechiae	**Blood sugar monitoring (at least twice):**
Sclerema	Hyperglycemia (>180 mg/dL)
**Respiratory Instability:**	Hypoglycemia (<45 mg/dL)
Apnea or	**Metabolic acidosis:**
Tachypnea or	Base deficit >10 mEq/L or
Increased oxygen demand or	Serum lactate >2 mmol/L
Increased need for ventilation support	
**Gastrointestinal:**	
Nutritional intolerance	
Insufficient breastfeeding	
Abdominal distension	
**Non-specific:**	
Irritability	
Lethargy	
Hypotonia	

**Table 2 cells-11-00192-t002:** Early onset neonatal sepsis (EONS) group compared to control group. Clinical characteristics compared between the EONS group and control group. *p*-values for gender, mortality, and positive blood culture were obtained by Fisher’s exact tests. Data are expressed as a percentage (%). For all other categories Mann–Whitney tests were used. These data are expressed as mean (SD). Significance was set at 0.05. n.s.—not significant.

	EONS (*n* = 3)	Controls (*n* = 10)	*p*
Age (d)	2 (1)	1.5 (0.71)	n.s.
Gestational age (weeks)	26.76 (0.78)	29.89 (4.53)	n.s.
Gender (female)	1/3 (33.33%)	3/10 (30%)	n.s.
Mortality	2/3 (66.67%)	0/10 (0%)	0.038
Positive blood culture	0/3 (0%)	0/10 (0%)	n.s.
Leucocytes (10^9^/L)	37.9 (31.77)	24.32 (22.97)	n.s.
Platelets (10^9^/L)	111.0 (64.09)	270.5 (78.85)	0.012
Blood gas analysis			
pH	7.204 (0.15)	7.313 (0.06)	n.s.
pO_2_ (mm/Hg)	41.07 (10.04)	32.07 (4.95)	n.s.
pCO_2_ (mm/Hg)	48.97 (9.96)	49.86 (11.03)	n.s.
Lactate (mmol/L)	11.27 (11.03)	2.44 (1.48)	0.023
Base deficit	−7.45 (7.99)	−2.171 (2.06)	n.s.
Glucose (mg/dL)	178.0 (36.59)	96.44 (42.21)	0.018

**Table 3 cells-11-00192-t003:** Late onset neonatal sepsis (LONS) group compared to control group. Clinical characteristics compared between the LONS group and control group. *p*-values for gender, mortality, and positive blood culture were obtained by Fisher’s exact tests. Data are expressed as a percentage (%). For all other categories, Mann–Whitney tests were used. These data are expressed as mean (SD). n.s.—not significant.

	LONS (*n* = 6)	Controls (*n* = 12)	*p*
Age (d)	44.01 (35.63)	25.58 (22.72)	n.s.
Gestational age (weeks)	26.52 (2.07)	28.99 (3.64)	n.s.
Gender (female)	2/6 (33.33%)	5/12 (41.67%)	n.s.
Mortality	1/6 (16.67%)	0/12 (0%)	n.s.
Positive blood culture	1/6 (16.67%)	0/12 (0%)	n.s.
Leucocytes (10^9^/L)	9.2 (2.63)	12.58 (5.87)	n.s.
Platelets (10^9^/L)	207.2 (150.5)	476.6 (140.5)	0.006
Blood gas analysis			
pH	7.314 (0.12)	7.340 (0.07)	n.s.
pO_2_ (mm/Hg)	36.82 (8.03)	33.9 (9.57)	n.s.
pCO_2_ (mm/Hg)	60.6 (20.81)	52.89 (9.98)	n.s.
Lactate (mmol/L)	2.38 (1.39)	1.48 (0.39)	n.s.
Base deficit	−1.45 (0.21)	−4.1 (3.31)	n.s.
Glucose (mg/dL)	163.2 (120.9)	97.89 (23.78)	n.s.

**Table 4 cells-11-00192-t004:** Detailed clinical information about all EMA positive participants (*n* = 9). All positive findings are marked (x). Sum of all positive items for clinical and laboratory findings were calculated. At least two clinical and two laboratory findings were needed to fulfill sepsis definition (EMA positive) [6].

No.	1	2	3	4	5	6	7	8	9
Study group	EONS	EONS	EONS	LONS	LONS	LONS	LONS	LONS	LONS
Mortality	x	x		x					
Proven infection				x					
Clinical:									
Body temperature		x					x		
Cardiovascular instability			x	x	x			x	x
Urine amount < 1 mL/kg/h	x					x			
Hypotension		x				x			
Impaired peripheral perfusion		x					x		
Petechiae									
Sclerema									
Respiratory instability		x	x	x	x	x	x	x	x
Nutritional intolerance	x							x	
Abdominal distension									
Irritability	x				x				
Lethargy		x						x	
Positive items	3	5	2	2	3	3	3	4	2
Laboratory:									
Leucocyte count	x	x							
Neutrophil ratio									
Platelet count	x	x		x					x
CRP				x	x	x	x	x	
Procalcitonin									
Hyperglycemia			x			x			x
Hypoglycemia	x								
Base deficit		x			x			x	
Serum lactate	x	x	x		x		x		x
Positive items	4	4	2	2	3	2	2	2	3
EMA positive	x	x	x	x	x	x	x	x	x

## Data Availability

Data is available upon request.

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
