# Peer review of "cfDNA and DNases: New Biomarkers of Sepsis in Preterm Neonates—A Pilot Study"

_cells, 2022, doi:10.3390/cells11020192_

Round 1
Reviewer 1 Report
Introduction
No comments
Methods
My main issue with the approach the authors have taken is the use of subgrouping using EMA criteria. My understanding is that the authors have generated a sepsis group based on EMA criteria-positive patients and a “control group” based on suspected sepsis patients that do not meet EMA criteria. As with any scoring system for sepsis, the EMA criteria is not perfect and would not rule out sepsis. It seems likely that many of the “controls” could still have sepsis and therefore may have elevated inflammatory markers including NETs and cfDNA. An appropriate control group for such a study should be preterm infants with no suspected sepsis/infection. If the authors had access to leftover blood from such a control group, I believe this could put their findings into context and may produce more meaningful findings.
Line 69: "rests of blood samples". Is this a typographical error? Should be "Tests"?
Can the authors clarify how the cohort was selected? Were patients selected prospectively or were there biobanked plasma samples that were selected retrospectively?
Were there other inclusion/exclusion criteria?
Can the authors clarify whether patients were selected based on suspected sepsis? It isn’t entirely clear.
Can the authors define the EMA sepsis scoring system further in methods e.g. in a table? What parameters are used?
Were there any restrictions on the time between collection time and centrifugation/freezing? Since these samples would have been used for diagnostic purposes first, what was the delay from collection to storage of plasma? If DNAse remains active in the blood post collection, then presumably NETS can be degraded more in samples that have had a longer time at room temperature. It would be helpful to know the median time and range of the time from collection to storage.
Can the authors provide more information on the sytox green cfDNA method?
Line 116: misuse of deducted. Should be "deduced"
Authors use t-tests to measure differences between groups. Was all data parametric? If so could the authors mention whether a normality test was used? Given the low sample size, I would be surprised if the data was normally distributed. Authors should consider this and reanalyse the data with a non-parametric test.
From the results it seems that there are 2 control groups, one for EONS and one for LONS, based on the time of suspected sepsis? This isn’t clear until one reaches the results section. Could the authors make this clear in the methods section.
A flow diagram would help to understand the patients recruited and the subgrouping.
Results:
Given that only 9 patients met EMA criteria, could the specific criteria that each patient was positive for be presented as supplementary results?
Can the authors include some description or summary of the data in tables 1 and 2?
Given that CRP is included in the EMA criteria, it isn't surprising that this was significantly elevated in the sepsis group. Should this be included in results if you were already selecting patients based on this? It seems redundant.
It seems from some of the graphs that there are negative values? The violin plot goes below 0. Can the authors review this?
Discussion
Line 196: “it appears 196 that we found a dysregulation of NET mediated immune response in neonatal sepsis”. I don’t agree with this statement. Given that the NET markers were not significantly different, the data actually suggests more that the cfDNA could result from a non-NET-mediated process, i.e. necrosis, other inflammatory processes. This is an interesting finding which could be hypothesis-generating, but the authors seem to focus more on the idea of NETs, which this data does not support. I think this paper could be improved by including more discussion on other possible sources of cfDNA.
I don’t think authors can justify claiming that the lack of significant differences in NETosis markers could be due to low sample size, but accept the significant results from the other markers without the same caveat. It should be an overall limitation that the sample size is low, not only in reference to the results that don’t support the authors’ hypothesis.
One other limitation to address would be the use of DNAse 1 level rather than paired with DNAse activity. It is possible that DNAse activity is decreased in the sepsis groups, rendering the DNAse protein increase meaningless.
More discussion relating to the control group is required. What does the control group represent? I.e. what are the other conditions in preterm infants that present as suspected sepsis? Could NETs or cfDNA be elevated in these conditions, given that these markers can occur in sterile inflammatory conditions?
Author Response
Reviewer 1
My main issue with the approach the authors have taken is the use of subgrouping using EMA criteria. My understanding is that the authors have generated a sepsis group based on EMA criteria-positive patients and a “control group” based on suspected sepsis patients that do not meet EMA criteria. As with any scoring system for sepsis, the EMA criteria is not perfect and would not rule out sepsis. It seems likely that many of the “controls” could still have sepsis and therefore may have elevated inflammatory markers including NETs and cfDNA. An appropriate control group for such a study should be preterm infants with no suspected sepsis/infection. If the authors had access to leftover blood from such a control group, I believe this could put their findings into context and may produce more meaningful findings.
Line 69: "rests of blood samples". Is this a typographical error? Should be "Tests"?
Corrrected
Can the authors clarify how the cohort was selected? Were patients selected prospectively or were there biobanked plasma samples that were selected retrospectively?Were there other inclusion/exclusion criteria?
We revised the method section for better expantation of inclusion and exclusion criteria.
Can the authors clarify whether patients were selected based on suspected sepsis? It isn’t entirely clear.
Correct. The manuscript was revised accordingly.
Can the authors define the EMA sepsis scoring system further in methods e.g. in a table? What parameters are used?
A table with the EMA sepsis score items was added, including a detailed chart about all EMA positive participants, which can be found in the supplemental material.
Were there any restrictions on the time between collection time and centrifugation/freezing? Since these samples would have been used for diagnostic purposes first, what was the delay from collection to storage of plasma? If DNAse remains active in the blood post collection, then presumably NETS can be degraded more in samples that have had a longer time at room temperature. It would be helpful to know the median time and range of the time from collection to storage.
Blood samples were processed immediately. This information was added.
Can the authors provide more information on the sytox green cfDNA method?
Information was added.
Line 116: misuse of deducted. Should be "deduced"
Corrected.
Authors use t-tests to measure differences between groups. Was all data parametric? If so could the authors mention whether a normality test was used? Given the low sample size, I would be surprised if the data was normally distributed. Authors should consider this and reanalyse the data with a non-parametric test.
We changed it to a non-parametric test. The manuscript as well as the figures were revised accordingly.
From the results it seems that there are 2 control groups, one for EONS and one for LONS, based on the time of suspected sepsis? This isn’t clear until one reaches the results section. Could the authors make this clear in the methods section.
Correct. Information was added.
A flow diagram would help to understand the patients recruited and the subgrouping.
The flow diagram was added.
Results:
Given that only 9 patients met EMA criteria, could the specific criteria that each patient was positive for be presented as supplementary results?
Supplement including these informations was added.
Can the authors include some description or summary of the data in tables 1 and 2?
Information was added.
Given that CRP is included in the EMA criteria, it isn't surprising that this was significantly elevated in the sepsis group. Should this be included in results if you were already selecting patients based on this? It seems redundant.
This was fact was addressed.
It seems from some of the graphs that there are negative values? The violin plot goes below 0. Can the authors review this?
We measured no negative data points, but some density plot reached negative values (mathematically).
Discussion
Line 196: “it appears 196 that we found a dysregulation of NET mediated immune response in neonatal sepsis”. I don’t agree with this statement. Given that the NET markers were not significantly different, the data actually suggests more that the cfDNA could result from a non-NET-mediated process, i.e. necrosis, other inflammatory processes. This is an interesting finding which could be hypothesis-generating, but the authors seem to focus more on the idea of NETs, which this data does not support. I think this paper could be improved by including more discussion on other possible sources of cfDNA.
Excellent point. We revised the manuscript accordingly.
I don’t think authors can justify claiming that the lack of significant differences in NETosis markers could be due to low sample size, but accept the significant results from the other markers without the same caveat. It should be an overall limitation that the sample size is low, not only in reference to the results that don’t support the authors’ hypothesis.
This was added to the limitation section.
One other limitation to address would be the use of DNAse 1 level rather than paired with DNAse activity. It is possible that DNAse activity is decreased in the sepsis groups, rendering the DNAse protein increase meaningless.
Unfortunately, due to the limited blood volume we were unable to perform a DNase activity assay. It is very good point. We added this to the discussion section.
More discussion relating to the control group is required. What does the control group represent? I.e. what are the other conditions in preterm infants that present as suspected sepsis? Could NETs or cfDNA be elevated in these conditions, given that these markers can occur in sterile inflammatory conditions?
The discussion was revised accordingly.
Reviewer 2 Report
In this study, Moritz Lenz and colleagues aim to evaluate the diagnostic value of NETs in sepsis diagnosis in neonatal preterm infants. Overall the manuscript is well written. The concept of the study is interesting. Looking for new biomarkers with high sensitivity and specificity is one of the main research fields in sepsis. However, the manuscript can be significantly improved, mainly by increasing the study group.
1) The main limitation of this study is the low sample size. Very small samples undermine the validity of the study.
2) The authors should describe the control group in more detail. Why does its number change from n=10 (Table 1) to n=12 (Table 2)?
3) The study is only a comparison between the test group and the control group. Statistical analysis should be extended.
Author Response
Reviewer 2
In this study, Moritz Lenz and colleagues aim to evaluate the diagnostic value of NETs in sepsis diagnosis in neonatal preterm infants. Overall the manuscript is well written. The concept of the study is interesting. Looking for new biomarkers with high sensitivity and specificity is one of the main research fields in sepsis. However, the manuscript can be significantly improved, mainly by increasing the study group.
1) The main limitation of this study is the low sample size. Very small samples undermine the validity of the study.
The limitation section was revised.
2) The authors should describe the control group in more detail. Why does its number change from n=10 (Table 1) to n=12 (Table 2)?
Information was added.
3) The study is only a comparison between the test group and the control group. Statistical analysis should be extended.
We concentrated on the comparison of sepsis vs. non-sepsis as we believe it the most relevant in a clinical setting.
Reviewer 3 Report
The authors report on NETs as a potential biomarker for the diagnosis of early and late onset neonatal sepsis in a cohort of 31 preterm neonates with suspected sepsis. They found that, compared to control groups, cfDNA, DNAse I, and CRP were significantly elevated in the 3 neonates with EONS and the 6 with LONS.
Of note, CRP was the variable that seemed to have the greatest statistical significance this pilot study.
Author Response
Reviewer 3
The authors report on NETs as a potential biomarker for the diagnosis of early and late onset neonatal sepsis in a cohort of 31 preterm neonates with suspected sepsis. They found that, compared to control groups, cfDNA, DNAse I, and CRP were significantly elevated in the 3 neonates with EONS and the 6 with LONS.
Of note, CRP was the variable that seemed to have the greatest statistical significance this pilot study.
The is no surprise as the EMA criteria which were used in this study include CRP. We revised the manuscript for better clarity.
Reviewer 4 Report
This report relates to plasma samples in newborn infants under conditions of neonatal sepsis. The authors speculate that products of NETs appear in plasma of infants with neonatal sepsis and seem to correlate with the clinical stats for these infants. It is suggested that activated PMNs in the blood form NETs, which are known to be highly prothrombotic and proinflammatory activities, are responsible for organ dysfunction in these infants. The basis for such conclusions is related to the presence of plasma markers suggestive of NETs. While the presence of these markers is supportive of such conclusions, it is important to consider that such conclusions are very provisional, requiring confirmation by additional studies of pre-term infants with sepsis. In Figures 1 and 2, the levels of cfDNA, nucleosomes, DNase-1 and CRP are statistically significant, but not for neutrophil elastase and H3Cit. The same is true in data in Figure 2: cfDNA, DNase-1 and CRP are statistically significantly elevated, but not for nucleosome, neutral elastase, and H3Cit. Another concern is the duration of methylation effects on the histones and the production of proinflammatory cytokines. If the cytokines are only transiently effective, there must be a mechanism of blocking the methyl group or the cytokines. Otherwise, there would be unmitigated production of damaging cytokines following methylation of the histones. This needs to be discussed.
This report is important in suggesting that plasma markers in plasma may indicate the presence of markers of NETs, suggesting that activation products of PMN NETs may be playing a role in newborn sepsis. However, additional studies should be done to confirm the data presented, and there should be analysis of other proinflammatory cytokines in plasma (e.g. TNF, IL-1β, IL-6, IL-17A-F, etc.) to determine if other cytokines in plasma may also be playing a role in neonatal sepsis.
Author Response
Thank you very much.
Round 2
Reviewer 2 Report
I think the response to the reviewers' comments and revision of the manuscripts are acceptable.
Author Response
Thank you very much.